# Assessment of Stakeholder’s Perceptions of the Value of Coral Reef Ecosystem Services: The Case of Gili Matra Marine Tourism Park

**DOI:** 10.3390/ijerph20010089

**Published:** 2022-12-21

**Authors:** Ratu Fathia Rahmadyani, Paul Dargusch, Luky Adrianto

**Affiliations:** 1School of Earth and Environmental Sciences; University of Queensland, St. Lucia, Brisbane, QLD 4072, Australia; 2Department of Geography, State University of Malang, Malang 65145, Indonesia; 3Faculty of Fisheries and Marine Sciences, IPB University, Bogor 16680, Indonesia; 4Center for Coastal and Marine Resources Studies, IPB University, Bogor 16680, Indonesia

**Keywords:** ecosystem services, coral reef, MPA, stakeholder, resources management, social perception, Gili Matra

## Abstract

Ecosystem services is a concept broadly applicable to describe environmental interrelations with human activities. It serves as a practical instrument for assessing the success of resource management in natural reserves, with the goals of maximising conservation effort and achieving sustainable use. The Gili Matra Marine Tourism Park (GMMTP) has been extensively researched as a marine protected area centred on anthropocentric activities of marine-based tourism. However, there still a lack of research to address the full scope of ecosystem services derived from the coral reef ecosystem. From an ecosystem services viewpoint, the study’s objectives were to define the services obtained from the GMMTP’s coral reef ecosystem, relevant stakeholders, and how their utilisation activities were posed as drivers of changes that reflect the flow of services and the possible implications of these. Marine tourism, capture fisheries, and land-based activities were identified as services impacting upon the regulating and supporting services, with the resultant compounding externalities potentially degrading the services’ utilisation value. Although there have been certain changes in community behaviour that may reduce the intensity of the impacts, the present prediction of service flow still confirms the previous statement. The results provided insight into current resources management implications on the state of ecosystem services. Overall, failing to recognise the causes that drives the interaction of these ecosystem services will increase the risk of incurring unexpected trade-offs, restricting the potential for resources’ synergies, and eventually causing drastic and irreversible changes in the provision of coral reef ecosystem services in the GMMTP.

## 1. Introduction

The notion of ecosystem services (ES) has become widely accepted as a proxy for integrating ecosystem and societal features [1]. Ecosystem services are described as the direct and indirect benefits obtained from ecosystems that contribute to the state of human wellbeing [2]. ES is commonly used as a proxy for environmental management assessment, as it portrays the most fundamental relationship and reciprocity between human society and the ecosystem [3]. Ultimately, it is widely established that human utilisation has negatively influenced the state of the world’s ES, reducing them to 60% of their original condition [2]. Therefore, the ES concept has been advocated as a unit to address sustainability challenges worldwide [4,5,6].

The coral reef ecosystem is renowned as one of the most productive and biologically diverse ecosystems globally [3,7]. Humans and coral reefs form such a complex relationship that it is characterised as a human-dominated ecosystem, most evident in islands where the landscapes are isolated and bordered by the ocean. Communities have depended on the coral reef and its services; as the utilisation practices grew more extensive, complex behaviours, such as overexploitation and destructive activities, also arose [2]. The Marine Protected Area (MPA) was a concept first introduced as an extension to the terrestrial protected area programs and has now become the most substantial part of global marine conservation efforts [8,9]. The established primary management objectives of an MPA are (i) the protection of a natural ecosystem; (ii) the management of habitat and species; and (iii) the sustainable use of natural resources [9].

First established in 1993, Gili Matra Marine Tourism Park (GMMTP) was one of the exemplary locations for good MPA practice in Indonesia [10]. Built around the core strength of its high coral reef biodiversity, the management of GMMTP aims to maximise the sustainable utilisation of the ocean resources through tourism. However, the rapid growth of anthropogenic activities and their dynamics amongst multi-users soon created severe consequences for the coral reef ecosystems within the GMMTP [11,12]. Many studies have analysed how the rapid growth of tourism on a mass scale has impacted the GMMTP coral reef in ecological, economic, and socio-cultural aspects [13,14,15,16]. 

However, the current available studies incorporating the ES concept as a utilitarian instrument to address GMMTP’s environmental issues were scarce, or focused only on a single predominant service. There was a lack of clear identification of the wide range of ES typologies that the GMMTP coral reef provides. This oversight may result in a management failure that can cause regime shifts and unexpected loss of multiple services, especially ones that have not been appropriately identified and addressed. Consequently, this failure would impact all the services’ societal benefits to the GMMTP community [17]. Hence, it is crucial to improve our broad understanding of coral reef services provision within the GMMTP.

Considering the nature of the GMMTP as a multi-user MPA, the interdependencies between the coral reef ecosystem and the resource users could unravel potential adaptation challenges to the community under fluctuating ES conditions [17,18]. Due to events such as the Lombok Major Earthquake in 2018 and the COVID-19 pandemic in 2020, the GMMTP community has undergone profound lifestyle alterations, which were later reflected in their interactions with ES provisions. A follow-up assessment of coral reef services must be conducted to maintain ecological and social resilience as the implied goal under MPA establishment [9,18]. 

Some critiques stated that ES assessment often lacks insight into the social dimension, resulting in an incomplete portrayal of ES values. In this context, the social dimension is defined as “how individuals, communities, and societies interact with, affect, and are affected by natural ecosystems and their changes through time” [1]. “Stakeholders” was a term used to describe users who interact with the ES daily; hence, their perspective may provide concise information on how these ES are valued [15,19]. 

This study identified the range of services provided by the coral reef ecosystem in the GMMTP and the associated stakeholders, as well as their form of interaction. Subsequently, these identification exercises were visualised using a spatial map to capture the ES supply, flow, and demand, as well as the interactions between each ES. The aim was to observe any stresses or negating responses that may result from ES interactions and the consequential externalities they may cause to the overall service provision and human wellbeing. 

This study is intended to provide supplemental ES knowledge on MPA management to improve social and conservation outcomes. This perspective would serve as important information to adaptational challenges in the face of ecosystem service changes. It also provides an opening for ecosystem resilience maintenance, which is the primary objective of an MPA. Additionally, understanding how the stakeholders interact with the ES and their dynamics may increase management benefits, by enhancing engagement to generate social change [1,19,20].

## 2. Materials and Methods

### 2.1. Study Area

The Gili Matra Marine Tourism Park is a National Marine Protected Area, located in the northern part of the Lombok Strait, extending from 116°01′34″ E to 116°05′18″ E and 8°20′02” S to 8°22′16” S (Figure 1.) It is administered under the West Nusa Tenggara Province, North Lombok District. The GMMTP was officially established as a protected area under the term “Marine Tourism Park” on 4 March 2009, through ministerial decree [11].

The GMMTP is formed from a cluster of three small islands (or “Gili” in the local language), namely Meno, Ayer, and Trawangan, shortened to “Matra”. The GMMTP is categorised as semi-open inner islands supported by a stable seafloor. These characteristics support abundant marine natural resources in the fringing reefs, consisting of a diversity of soft and hard coral populations surrounding the islands. The Ministry of Marine Affairs and Fisheries data show that the GMMTP coral reef houses up to 40 genera of hard corals and 344 species of reef fish. The GMMTP waters also foster iconic marine species, such as sharks, rays and sea turtles [11,21]. 

The management plan for the GMMTP for 2014–2034 was established under the Ministerial Decree Number 57 Year 2014. The management plan covers some 2273.56 ha of the GMMTP waters. The management plant allocated seven spatial zones within the GMMTP waters, based on their ecological, social, and economic potential. The zones comprise of Core Zone (Zona Inti), Sustainable Fisheries Zone (Zona Perikanan Berkelanjutan), Sustainable Reef Fisheries Sub-zone (Sub-zona Perikanan Berkelanjutan Karang), Utilisation Zone (Zona Pemanfaatan), Protection Zone (Zona Perlindungan), Rehabilitation Zone (Zona Rehabilitasi) and Harbor Zone (Zona Pelabuhan). Each zone was delineated for specific purposes, and allowable activities within the area were regulated [11].

The Core Zone, which encompasses 94.81 ha of the waters, is a no-enter zone specifically set for conserving marine habitats and populations. Except for research or educational purposes, no utilisation or extraction activities were allowed in the zone. The Sustainable Fisheries Zone, which also includes the Sustainable reef fisheries Sub-zone, comprising 1870.1 ha of the area, was open for utilisation, specifically for small-scale fisheries activities, using sustainable fishing and traditional gear to support the development of culture-based recreational fisheries. The Utilisation Zone, comprising 207.49 ha, was an open utilisation area, intended to support the development of marine-based tourism, such as scuba diving, snorkelling, kayaking and glass-bottom boat cruises. The Protection Zone covered 7.44 Ha and was purposed to protect critical habitats, notably the Blue coral colonies (*Heliopora* sp.) that occupy the shallow waters of GMMTP. The Rehabilitation Zone consisted of 36.63 ha, designated for coral reef and seagrass rehabilitation efforts. Lastly, the Harbor Zone covered 61.64 ha and was designated for ships and their mooring. All destructive fishing gear was prohibited in all areas, as well as aquaculture and mooring anchors (except in the Harbor Zone), to protect the coral reef ecosystem of the GMMTP [11]. 

GMMTP is a globally attractive marine tourism destination, bringing about 500,000 tourists yearly. Tourists purchase an entry ticket to conduct activities in the waters of GMMTP. Revenue from tickets alone is estimated at around three billion Indonesia Rupiah, equivalent to USD 200,000, per year. Tourists comprise local and international tourists [22]. The tourism revenue of GMMTP contributes up to 70% of the North Lombok economy. More than 50% of the Gili Matra population works in activities related to the tourism industry. In addition, it also creates substantial employment opportunities for mainland communities (North Lombok District) [14]. 

### 2.2. Methodology

#### 2.2.1. Ecosystem Services and Stakeholders Identification

The identification of coral reef ES and the related stakeholders in the GMMTP was conducted through a desk study. Data sources used were past literature with a spatial focus on GMMTP, the zonation and management plan of GMMTP decreed under the Indonesian ministry of marine affairs and fisheries, and other relevant public data acquired from related Indonesian agencies or institutions [11,23]. Subsequently, we used expert judgment to verify the desk study results. In this context, experts are individuals with advanced knowledge regarding the GMMTP and the surrounding coral reef ecosystem. We contacted the GMMTP local community, government institutions managing the GMMTP, and local universities with frequent collaborations with the GMMTP. In total, three experts were selected to participate, with varying backgrounds of locality, profession and level of education, to capture the diversity of opinion. They are referred hereto as “experts”. To verify the ES and stakeholder identification result, these experts were given a set of open-ended questions, adapted and modified from a similar study by Aziz et al. (2016) [23]:What activities interact/derive benefits, whether directly or indirectly, from the coral reef ecosystems?Were these activities of an extractive/non-extractive nature?Who conducted these activities?What other industries/business interacts with the coral reef ecosystem services?

Next, a similar exercise was performed, identifying the stakeholder groups related to the coral reef ES by answering an additional set of questions:Who makes the decisions related to these ecosystem services?Who has the responsibility for the benefit and management of these ecosystem services?

Additionally, the respective interests of the stakeholder groups and activities related to ecosystem services utilisation were noted [23]. The broad identification results of the coral reef ES and related stakeholders were then constructed into a compilation table. 

We then performed a screening exercise on the list of broadly identified stakeholder groups. We believe that each stakeholder group differs in their level of interaction with the coral reef ecosystem. Stakeholders who participate most in actions that directly influence the coral reef ES will subsequently be the drivers of change in the ES flow cycle [1]. Their knowledge was more likely to portray the actual value of the ES in the area, which makes them key stakeholders. Experts were asked to answer Likert-scale questions to determine the stakeholders’ level of relevance and influence over the identified ecosystem services, ranging from “Not relevant/influential at all” (= 1) to “Very relevant/influential” (= 4) (see Appendix A). Relevance refers to the significance of the coral reef ES to each stakeholder group, whereas influence refers to the power each stakeholder group holds over the outcomes (benefits, rights, access, decisions) of the ES [23]. The intersections of these values reflected their priority level in coral reef ES utilisation. The results were incorporated into a four-quadrant matrix. The stakeholders are categorised into four types: 

(1) The low relevance–low influence group characterises stakeholders with scarce interaction with the coral reef ecosystem and whose actions cannot directly affect the coral reef of the GMMTP;

(2) The high relevance-low influence group characterises stakeholders with frequent interaction with the coral reef ecosystem but who are not involved in actions that directly affect the coral reef of the GMMTP;

(3) The low relevance-high influence group characterises stakeholders with scarce interaction with the coral reef ecosystem but who may be involved in actions that directly affect the coral reef of the GMMTP; and

(4) The high importance-high influence group, which characterises stakeholders with frequent interaction with the coral reef ecosystem and involvement in actions that directly affect the coral reef of the GMMTP (Figure 2) [1]. 

The high importance-high influence group (Category 4) serves as the ‘key stakeholder’, which is the primary focus for the following stage of exercises (Figure 2).

#### 2.2.2. Ecosystem Services Valuation and Mapping

Once the ecosystem services and key stakeholders were identified, we carried out interviews with the representatives of the identified key stakeholder groups, to figure out the ecosystem services that drive the ecological condition of the coral reef ecosystem, as well as the social wellbeing of the GMMTP community. A set of interview questions was developed to measure the respondents’ assigned values of importance (ES’ contribution to human wellbeing); vulnerability (current exposure to threats that would lead ES to be degraded or lost); dependency (how the ES assist their daily lives and livelihood); and preference (level of priority) upon the ES. The assigned scores ranged from 1 to 5, with 1 being the lowest and 5 the highest, depending on the context [1] (see Appendix A). The results were then entered into a four-quadrant matrix. The importance–vulnerability matrix portrayed the criticality of the ES, whereas dependence–preference portrayed the prioritisation of ES. The services which attain high scores in importance–vulnerability values portrayed a highly critical ES and can be observed in the upper right quadrant of the matrix. At the same time, high scoring in the dependence–preference values signifies a high priority level of the ES. Combining the result of the scorings, the ES perceived as highly critical and prioritized in the upper right quadrant of both matrices were identified as the key ecosystem services in the GMMTP coral reef.

Based on the valuation results, key ES of the coral reef in the GMMTP were illustrated on a spatial map. Spatial visualisation aims to capture the delivery flow from each ES and simplify the observation of relationships between ES, rather than just focusing on the theoretical foundation of biophysical interactions [23]. The identification of each spatial area is mainly derived from identified ES zones in the Management and Zoning Plan for the Gili Matra Marine Tourism Park 2014–2034 (KEPMEN-KP No.57 Year 2014), supplementary literature [11,13,21,23], and communications with key stakeholders. 

#### 2.2.3. Stakeholder’s Analysis of the Coral Reef Ecosystem Services

Additionally, stakeholder analysis was performed. Stakeholder analysis is a system for gathering information about groups or persons affected by a topic, categorising it, describing different forms of relationships between groups and areas, and exploring trade-offs when possible [24]. During the interview sessions, respondents were asked several exploratory questions regarding stakeholders’ synergisms, to determine the distribution and provision of ecosystem services. The designed questions were adapted from a similar study by Aziz et al. (2016) [23]: What is the level of influence of the stakeholders?Who has direct or indirect impacts on the ecosystem services?How would the stakeholders’ activities be affected if all activities were conducted simultaneously?What is their willingness and capacity to participate in ES management?What are the current and future interests of the various stakeholders in the use and management of the ES?What are the social and environmental impacts, both positive and negative, of the past and current uses of ecosystem services by the stakeholders?

The information gathered from the interviews was then incorporated using the identification, valuation and mapping exercises to construct a concept of the systemic interactions between each ES, drivers of ES, the direction in which the drivers were moving, and trade-off projections [1,24,25,26]. The results were combined into a conceptual diagram with a simplistic, understandable flow. Ultimately, the diagram evaluates the management implementation outcome in the GMMTP.

## 3. Results & Discussion

### 3.1. Gili Matra Marine Tourism Park Coral Reef Ecosystem Services and Stakeholders Mapping

#### 3.1.1. Ecosystem Services Identification in the GMMTP Coral Reef

We present the results of ES identification in the compilation table below (Table 1). The identification was conducted by studying planning documents, existing reports, and scientific literature [11,14,27,28]. The results were then examined by experts with specific familiarity of the GMMTP area, to ensure that the identified ES were relevant. The identified ecosystem services were categorised into four typologies, based on the Millennium Ecosystem Assessment’s classification [4]. Following the method by Aziz et al. (2016) [22], these identified services were further itemised based on its services type, service description, related stakeholder groups, and the scale in which the benefits of the services outreached. For example, local benefits indicate that the services’ products were mainly distributed to communities within the local scope (GMMTP and surrounding islands) reciprocally for international benefits, whereas general benefits indicate that the services were distributed indistinctly. 

Capture fisheries serve as the primary provisioning service for the GMMTP community. Fisheries activity by the local community of GMMTP is predominantly small scale. Vessel size is not larger than 40 hp, with the main fishing gear being handlines, spearguns, and gill nets [11,29]. The fisheries activity consists of 1-day fishing, with a catch quota of up to 100 kg/trip. There are no designated fish landing facilities in the GMMTP islands, so the catch is distributed to collectors or traded locally within the three islands. Like many characteristics of small islands, capture fisheries serve as the primary source of protein intake for the community and the main provider for tourism activities. Since commercialism grew, fisheries activities in GMMTP have been gradually replaced by tourism, and currently only contribute up to 16% of the GMMTP community income [11,13,30]. The processed seafood industry is another subsidiary product currently developed as an alternative livelihood scheme for the community. The industry is managed by the women group of the GMMTP. Although the activity is still under development at the time of the data collection, it was recognised as a crucial service, as it is expected to incentivise the development of specialised local products and gender involvement [29].

The cultural service provides non-material advantages, such as educational and tourism/recreation benefits; the latter serves as the backbone of GMMTP’s economy [13,31]. The predominant tourism activities in the coral reef area include diving, snorkelling, and canoeing, as well as recreational fishing, which operates on a smaller scale, considering the lesser demand. GMMTP is famously known for its underwater tourism (scuba diving and snorkelling), with 97% of incoming tourists estimated to participate in two underwater excursions. This identification revealed that snorkelling operators and glass-bottom boat operators were usually owned and operated jointly; hence, for simplicity, they were merged into one group. However, the number of these service operators may have decreased up to 30% due to the COVID-19 pandemic. 

Another form of major tourism service in the GMMTP area is beach tourism, which includes cafés, restaurants, shops, hotels, sunbathing spots, and more. The involvement of land activities in the dynamic of coral reef ecosystem services initially did not surface during the literature review, nor the experts’ discussion for ES identification, as they were not assessed as key ES related to the coral reef ecosystem. As a result, land activities were not discussed in the identification or valuation exercises. However, while conceptualising the ES interaction diagram, it was noted that land activities were closely associated with the coral reef’s function as a coastal protector. This was specifically related to shoreline tourism developments, where sand reclamation for beach leisure and hostel developments on the beachfront were flourishing [13,14]. To address this limitation, we included land activities’ impact and role in the discussion. It is noted that land activities were not recognised as a key ES of the coral reef ecosystem, but, rather, as an integral part of pressure drivers, highlighting the possible trade-offs and including them in our analysis.

The coral reef provides a regulating service by forming reefing structures that serve as physical barriers to protect the shoreline from tidal surges and extreme weather events, and contribute sediment input for land accretion [27,28,29]. The fringing reef ecosystem surrounding the GMMTP stabilises and protects the islands from extreme waves and current events [13,32]. Additionally, the natural physical breakdown of calcified coral due to wave action contributes to beach accretion through a hydrodynamical process [27]. This specific role of coral reef is crucial, especially in small islands such as GMMTP, where any disturbance to this function will result in an immediate impact, due to the isolated nature of the ecosystem. Evidently, several parts of the GMMTP have been experiencing severe erosion due to coral reef degradation [13,33,34].

The last essential ES provided by the coral reef is the biodiversity benefit (supporting service). The coral reef is amongst the ecosystems with the highest biodiversity in the world, and serves as a nursery, habitat and feeding ground for marine biota [7,27]. GMMTP’s biodiversity value is prominently known and is the main reason for its high tourism appeal. This biodiversity benefit is essential to delivering other ecosystem services in the GMMTP coral reef area.

#### 3.1.2. Stakeholders of the GMMTP Coral Reef Ecosystem Services

In the stakeholder identification exercise, we identified the stakeholders involved (both direct and indirectly) in all the coral reef ES in the GMMTP area. Each stakeholder group was then linked to the ES based on interaction (Table 1). The form of interaction falls into three categories: Managing agencies are stakeholders with jurisdiction to regulate or manage the ES;Community are stakeholders who benefit from the ES, both from the socio-economic and general perspective;Private businesses are businesses that derive significant economic yield from interaction (direct or indirect) with the ES [14,22,35].

Next, we conducted a stakeholder prioritisation exercise, to find key stakeholders based on their relevancy and influence over the ES [1]. The stakeholder prioritisation results are presented in the matrix below (Figure 3). The first type of low relevance–low influence stakeholder group consists of the Women Group, Youth Community, Cart and Bicycle Rentals, and the District Government. The Women Group, Youth Community, and Cart and Bicycle Rentals may be ranked as such due to their indirect influences and limited participation in coral reef ecosystem-related activities. 

The District Government was classified in the low relevance–low influence group, despite its jurisdiction over the governed region and resources. It was noted that there is a segregation over spatial jurisdiction, where the District Government governs the land, while BKKPN Kupang governs the waters. However, this may also signify the perceived incoordination between agencies over land and marine resources, which is crucial in landscapes such as small islands.

Based on the prioritisation in Figure 3, key stakeholders include governmental agencies with direct authority over the GMMTP’s coral reef ecosystem, i.e., BKKPN Kupang, Department of Tourism, Department of Forestry, Department of Fisheries, and Village government; businesses that derived direct profit from the delivery of coral reef ES, i.e., dive centres, snorkelling and glass boat operators; and communities/organisations with direct intervention and initiatives involving coral reef ecosystems, i.e., local communities, NGO and academia. Their roles are described in Table 2. BKKPN Kupang, Village Government, dive centres, snorkelling and boat operators, and conservation community emerged as the key actors. Based on our verification with the stakeholders, BKKPN Kupang holds the central position as the main body of authority for the MPA.

We conducted an ES appraisal with representatives of all 13 groups of identified key stakeholders (Table 2) to understand, from a social perspective, the ES state, level of priority, and relationships between each ES and stakeholder group involved. We believe that the knowledge of actors who possess high power and participate in direct interventions in the coral reef ecosystem would closely reflect the actual value of the resource services [35]. We contacted each of the identified key stakeholder groups in Table 2 to request an interview. Sometimes, one group was represented by two people, per the stakeholders’ request, and the interview would be run separately. Each participant comes from the diverse local, researcher, activist, and civil servant communities who live or operate in the GMMTP daily. We conducted the interviews until we reached data saturation, and no new information emerged. In total, 16 people were interviewed.

#### 3.1.3. Stakeholder Valuation of the GMMTP’s Coral Reef Ecosystem Services

The 16 participants were asked to score each coral reef ES based on their knowledge and representing the views of their stakeholder group. The results are presented in four-quadrant matrices (Figure 4). The stakeholders appraised most of the coral reef ES as critical (high importance—high vulnerability), except for the processed seafood and educational benefit. The low values for the processed seafood product were due to the initiative still being under development at the time of data collection. While the educational value scored low in criticality (Figure 4A), it scored higher in priority (Figure 4B). This captures the community’s high expectation for scientific enquiry. From discussions with the community, there appears to be a gap in dissemination of research results to the GMMTP community. Research activities outside those of government initiatives are frequently not reported to BKKPN Kupang as the principal custodian of the GMMTP. This fragmentation of findings may lead to the loss of valuable information crucial to the future management of GMMTP. Based on the exclusion of both these services from the key ES category, they were not included in further discussion of this paper, although it should be noted that their recognition during the identification stage presents them as potential subjects for further studies. 

Cultural services, specifically underwater tourist activities like diving and snorkelling, were valued highest in criticality (Figure 4A). Another service within the category is recreational fishing. However, it was valued much lower in criticality and priority, mostly due to lower demand, as it is largely restricted to the locals and domestic tourists [29,33]. Tourism is the backbone of GMMTP’s economy, supporting 80% of the community’s livelihood [33]. It is also the core focus on which the marine protected area is developed [11]. The high vulnerability score reflects the depreciation of the industry since the occurrence of the Lombok major earthquake in 2018. Based on the information from participants, the tourism industry was just starting to rebound following the earthquake when the COVID-19 outbreak occurred in 2020. Restriction on international and national travel subsequently shut down all tourism activities within GMMTP. This chain of events significantly affected the economy of the GMMTP community and, consequently, extended to the district level. A substantial drop was observed in the North Lombok economic growth in 2020, which corresponds to the loss of tourism income from the GMMTP [36]. The stakeholders believe that tourism is crucial to revitalising the GMMTP economy. This expectation was reflected in its high prioritisation score (Figure 4B). 

The supporting service’s criticality score was high (Figure 4A). However, its value in the vulnerability axis was nearer to neutral. These types of indirect functions of the coral reef are not outwardly observable [37]. Hence, they are often undervalued; however, it was scored highest in priority (Figure 4B). Most stakeholders recognised the importance of the coral reef ecosystem and how ecosystem degradation has caused adverse impacts on biodiversity and ecosystem quality. The coral reef functions as a habitat, nursery, and feeding ground, supporting the spillover of fisheries stocks to surrounding waters [38].

Moreover, the coral reef harbours cultural amenity of biodiversity appreciation, which is one of the tourism appeals of GMMTP [22]. For many of the biota, the coral reef serves as a nursery ground and is crucial to their early life cycle. Any disturbance to this function will directly affect species’ survival [11]. For example, GMMTP waters are well-known as a habitat of sea turtles, one of the most severely threatened marine species on earth. Three known sea turtle species exist within GMMTP waters: Hawksbill turtle, Green turtle, and Olive Ridley turtle. All are listed as protected species under CITES and protected under Indonesian law [21,39]. Besides sea turtles, several exotic marine species, such as sharks and rays, can be found in several parts of GMMTP waters. From an ecological perspective, coral reef degradation affects the quality of feeding ground, contributing to decrease in marine species populations. A study by Jupri et al. (2020) supported this notion by stating that fewer sea turtles were spotted, consistent with the rate of coral reef ecosystem decline. In addition, several turtle nesting points were recorded on the GMMTP sandy coast. A recent study found that overwash events due to coastal abrasion has impacted sea turtles’ nesting points [40].

The regulating service of coral reefs as coastal protector and shoreline stabiliser scored high in both criticality and priority (Figure 4A,B). The erosion threat in GMMTP has put the service into the community’s focus. Sand erosion has caused loss of some tourist amenities, such as sunbathing spots and snorkelling sites [41]. Erosion has also caused overwash and inundation, which may affect property areas and public roads, as evident in many instances [8]. In one case, it even affected a temporary waste shelter, which disrupted the waste processing chain within the island. 

Capture fisheries as a provisioning service scored high for criticality (Figure 4A) but comparatively lower for priority (Figure 4B). The high criticality score reflects capture fisheries as another major source of income for the community, next to tourism, and it holds high economic value. Demographical data from 2014 stated that the fisheries industry only represents 9% of the livelihood source in the GMMTP community [15]. However, based on the information from the community, the numbers have since increased significantly, due to the impact of the COVID-19 pandemic. Restrictive travel regulations have caused many tourism businesses within the area to collapse, which triggered most GMMTP locals to resort to fisheries as their primary livelihood source. This is indicated by the rise in number of fishermen from 2019 to 2020 (aggregate data of North Lombok district) [42]. Based on our discussion with the participants, the lower priority score may be the effect of satiety of resources. There was lower competition for resource use in the GMMTP fishing ground without other utilisation activities occurring during the COVID-19 pandemic. Fisheries activities were more prevalent, as waters were not restricted by other activities. However, criticality is still high in score, as the GMMTP’s fishermen voiced concerns about the increase in destructive fishing, low catch biomass, and the need to fish beyond the MPA boundary to fulfil catch quotas. 

#### 3.1.4. The GMMTP Coral Reef Ecosystem Services Map

Key ES in the GMMTP’s coral reef were segregated into five individual maps for visualisation. The spatial mapping for coral reef ES is primarily based on the spatial zones established in the Management and Zoning Plan for the Gili Matra Marine Tourism Park 2014–2034 (KEPMEN-KP No.57 Year 2014) [11]. The seven zones are each defined based on their ecological, social, and economic potential. It is therefore assumed that each of these zones has the highest potential to deliver specific ecosystem services and is the area where people generate impact or are impacted by ecosystem services. Further, we conducted ground-truthing by consulting the location of each ecosystem service with the participants of this study. Marine tourism activities are allowable in almost 92% of the GMMTP water area. Designated areas for snorkelling only take up 36% and are located in the shallow waters or nearshore parts, for safety purposes. Recreational fishing by speargun is regulated in certain territories, which prohibits catching fish other than by the traditional method. However, marine tourism activities make up the major service in the GMMTP waters. During peak season, tourists can number up to 61,000 per month [11,43,44]. Underwater tourism, such as snorkelling and scuba diving, was concentrated in 26 areas characterised by good coral cover, fish biomass, and exotic marine fauna [11]. Given that environment’s aesthetic appeal usually drives these types of underwater excursions, it does not generate extensive pressure on the supporting and regulating services of the coral reef. However, the issue arises from the scale on which the activities were conducted over a long period of time [11,43,44]. 

Capture fisheries cover around 82% of the GMMTP area. Approximately 28% of that area was specifically assigned for traditional fishing, defined as fishing activities specifically by traditional/local fishermen using traditional fishing gear or facilities [11]. Like marine tourism, fishermen will gather in locations with the highest potential catch, which causes the activity to largely overlap with marine tourism. As a result, conflict for resource appropriations between marine tourism and fisheries in GMMTP often arose [11,41]. Aside from that, capture fisheries generate direct pressure on the supporting and regulating services of the coral reef through destructive fishing practices, e.g., blasting, potassium cyanide, and use of Muroami nets. These illegal practices have stopped since the revival of *awig-awig* (local rule) in 2000 and the subsequent enactment of IUU fishing regulations [11]. However, violations are still found at present. 

The supporting service area was characterised by dedicated zones within the GMMTP, which serve as a sanctuary for fish and habitat, i.e., the core zone, rehabilitation zone, and protection zone. These areas hold critical supporting service values, including (1) conserving a representative sample of the natural landscape, (2) protecting sites critical for species reproduction, and (3) providing sites for species growth and settlement to generate a spillover effect to adjacent areas [45]. The area was also characterised by sites identified as settlements for exotic species, such as the blue coral, sharks, rays, and sea turtles [11,21]. The area makes up around 6% of the total area of the GMMTP, with a minimal to zero utilisation rate.

The regulating services are represented by the reefing coral structure that forms a physical barrier to protect the Gili coastlines from extreme events and wave-related disasters. The structural complexity of coral reefs serves as a foundation to sustain the biodiversity function of supporting services [41]. The coastlines are also included in this spatial visualisation, considering their interrelation in maintaining shore sediment through a hydrodynamic process [13,27,46,47]. 

The spatial observation through ES mapping reveals many overlapping spaces between ES utilisation activities, which can be perceived as negative drivers produced by mutually exclusive actions. To give an example, although marine tourism areas designate a large expanse of 92% of the GMMTP waters (Figure 5A, light blue shade), the actual activities are concentrated in specific areas due to several factors (high biodiversity, exotic species encounters, accessibility). Hence, in some instances, these activities overlap with sanctuary areas (Figure 5D) or areas that support species survival (mating grounds, migration areas). Another example is the overlap between capture fisheries areas (Figure 5B), marine tourism areas (Figure 5A) and sanctuary areas (Figure 5D). Although the designated area covers 82% of the GMMTP area, capture fisheries activities are concentrated in areas with the highest potential catch, more often located outside its designated zone that overlaps with marine tourism areas and sanctuary areas. The lack of a physical border between zones is often the cause of these violations [22].

Land activities, which were previously established as an integral part of the pressure driver to the coral reef ES dynamics in the GMMTP, are found in major parts of the GMMTP’s coastlines, which are covered with developments such as resorts, kiosks, and reclamation beach for leisure activities. These changes in the coastline due to increasing tourism demand has been documented to create erosion in several parts of the GMMTP coastlines [13]. These activities and their subsequent impact spatially collide with the regulating service of the coral reef as a coastal protector. This inference is based upon instances where sand and corals were mined for construction purposes and reclamation for sand leisure. Moreover, based on communications with the participants, illegal boat traffic for loading and unloading construction materials was often stated as the cause of coral reef harm around the island. These spatial mismatches present an important challenge for future conservation and management focus, especially in managing emerging trade-offs, should continuous degradation occur [17].

### 3.2. Stakeholder’s Perception of the Coral Reef Ecosystem Services

A simple conceptual relationship diagram between the identified coral reef ES in the GMMTP was constructed (Figure 6). The relationship diagram focuses on the negative drivers (negative change one ES may generate to one or multiple ES) [17]. This allows for identifying different drivers of ecosystem services decline and determining areas that would benefit from strategic management interventions. The analysis was presented from the perspective of BKKPN Kupang, to inform future management recommendations that are capturable for the principal custodian. The pressure on the ES was categorize as arrows. The head of the arrows points towards the direction in which the negative pressures are moving. The pressures are categorized into three types:

Direct pressure (impact generated from actions occurring inside the GMTTP and under the direct jurisdiction of the principal custodian);External pressure (impact generated from activities occurring inside or near the boundaries of GMTTP that falls outside the jurisdiction of the principal custodian); andCompounding pressure (the cascading impact originating from a pressured ES that leads to the decline of other services).

The ES relationship portrayed in this analysis was based on fundamental linkages, spatial observation, and information gathered from stakeholders [22].

Tourism and capture fisheries services were identified as activities with direct pressure on regulating and supporting ES, as they were under the immediate jurisdiction of BKKPN Kupang. Tourism generates IDR 1,102,165,479/year and is assumed to grow exponentially each year [16]. The negative impact of tourism on the biophysical capacity of the coral reef ecosystem in the GMMTP has been widely discussed in the science community [13,14,44,48,49]. Some coral reef areas with high biodiversity values in GMMTP are threatened with decline due to tourism overcapacity. At times, as much as 100 tourists may gather in the same location [45,50]. Lack of education briefing also resulted in damaging behaviour by tourists, whether intentional or unintentional [50,51,52,53]. Moreover, mooring anchors are widely responsible for the structural damage to coral reefs [54]. Observations in coral rehabilitation sites near popular tourism locations supported the notion of coral recruitment inhibition due to anthropogenic disturbances [54,55].

Capture fisheries were conducted at a smaller scale, generating an estimated IDR 151,130,418/ha/year [16]. Naturally, extractive activities hold the potential to impact the coral reef supporting services. The recent discovery of low fish biomass was likely associated with population collapse due to past overexploitation. In addition, destructive practices, such as blast fishing, have damaged the coral structure and affected the provision of regulating services [41,54,56]. As previously mentioned, illegal practices have since stopped, aided by the growing awareness of local fishermen. However, our discussion noted an increased risk of overexploitation and recurrence of violations as the demand for fisheries increases (due to the COVID-19 gap period). From the information gathered, violations tend to come from people outside the GMMTP local community. Monitoring and imposing sanctions should contribute to mitigating these pressures. However, reactions from authority were perceived to be slow and partial. BKKPN Kupang, as the key authoritative stakeholder in GMMTP, only possesses the mandate to monitor, whereas the enforcement of sanctions and further actions was a responsibility shared with other agencies. The community also contributes to the implementation of a zoning plan through the formation of surveillance groups; however, without adequate resources and capacity to carry out surveillance, their roles were limited to persuasion and verbal warnings to perpetrators. As a result, institutional organisations remain an inhibiting factor to management optimisation [10].

Secondarily, marine tourism promotes land-based activities that contribute to anthropogenic pressures on coral reefs in the form of land-waste accumulation, pollution run-off, increased boat traffic, and coastal developments [23,28]. Land activities were overlooked during the ES identification and valuation steps; however, it was established that these activities are closely associated with the coral reef ES’ interaction and are an integral part of the discussion [13,14]. Here, land activities were recognised as an external pressure on the regulating service, as it was not included under coral reef key ES; at the same time, its custody falls under the local government’s jurisdiction and not under the principal custodian. Observation from the field revealed that regulatory coordination between agencies was complex and bureaucratic. Until recently, land development in the GMMTP had been poorly regulated. As a result, developments often fail to adhere to good spatial management and, in some instances, even directly affect MPA implementation.

The function of coral reef as a coastal barrier was valued at IDR 9,569,065,000/year [16]. Past developments that allowed destructive activities, such as sand mining and use of coral blocks (limestone) as building materials, were reported as one of the main causes of GMMTP coral reef degradation, resulting in coastal erosion in the three Gilis [13,57]. Furthermore, unclear responsibility in managing the erosion on the three islands’ shores resulted in fragmented, ineffective results [13,14,45]. As one community member observed in Gili Air: “The protected parts [of the shore] were undisturbed, but the exposed parts [parts not protected by seawall] were eroding pretty severely. It was not like that previously.” Erosion events also directly affected GMMTP biodiversity, such as sea turtles, which lost their beach nesting sites due to changing beach morphology [40].

The interaction between marine tourism and capture fisheries was diagrammed to be bidirectional, i.e., affecting each other. In this case, optimising one of the services may cause the other service’s function to diminish [17]. Undeniably, community reliance on tourism was deep-rooted, and the realistic option for management was to build around the core strength of the GMMTP economy [57]. However, a gradual loss of the capture fishery industry due to conversion to tourism would be inevitable and would subsequently cause a collapse in the supply chain of the islands’ main food source. The GMMTP community would have to rely on transported goods from the mainland, exacerbating the current high commodity prices.

On the other hand, the overlap of fishing grounds with underwater tourism sites will decrease tourism satisfaction value, which will impact tourism demand and income [58]. The community has established a local rule called *awig-awig*, wherein resource use is implemented on a first-come, first-served basis. However, there were instances where it did not yield a good outcome, as fishermen often feel marginalised, due to the negative connotation of extractive use [15,57,59].

The combination of direct and external pressures on the regulating services is assumed to create compounding pressure on the supporting services, adversely affecting the provision of other ES. This cycle revealed that the continued increase in pressures would eventually degrade the quality of the ‘pressuring’ activities themselves (Figure 3). The compounding impact from the lost structure of coral reef will result in lost habitat, nursery ground, and feeding ground for associated marine biodiversity. Due to the degradation of coral reef ecosystem cover, symptoms of biodiversity decline, such as decreased fish biomass and target fish populations, and reduced encounters with exotic marine biota that characterise GMMTP were indicated several times in this study. The biodiversity value of coral reef ecosystem in the GMMTP was estimated at Rp10.821.883.500 [16]. In effect, the capture fisheries industry will experience a decline in fisheries yield [60]. Based on the discussion with a community member, this may trigger damaging behaviour from fishermen, using destructive means, such as blasting and poisoning, or jeopardising human safety, e.g., diving with a compressor. Loss of biodiversity will considerably impact tourism satisfaction, the main building block of marine tourism market price [58]. The loss of coastal protection will also affect many establishments built along the green belt exposed to erosion-related events, such as inundation and overwash. Approximately 5.05 ha, 1.79 ha, and 1.08 ha of shoreline area were lost due to erosion in Gili Air, Meno and Trawangan, respectively [2]. Abrasion threats will further cause a decline in property value, lost assets, income and jobs, not to mention the loss of public facilities, such as roads and the waste centre. The failure to mitigate the threat early on will result in astronomical mitigation and restoration costs.

## 4. Conclusions

Ecosystem services knowledge offers a simple perspective on an ecosystem’s social, economic and ecological value and associated functions. This study tries to identify the ecosystem services management gap by understanding its typologies, spatial patterns, and interactions. These may help to distinguish leverage points in which management investments can be applied to yield maximum benefit, without overlooking possible trade-offs.

Our study identifies the wide range of ecosystem services the coral reef ecosystem of the GMMTP provides, i.e., provisioning services (capture fisheries, processed seafood products); cultural services (marine tourism, educational benefit); supporting services (biodiversity benefit), and regulating services (coastal protection). The key stakeholders related to the coral reef ecosystem services fall under three categories: managing agencies, communities, and private businesses. BKKPN Kupang emerged as a central actor in managing the coral reef ecosystem in the GMMTP, considering the GMMTP’s status as an MPA. Based on the stakeholders’ perception, there seems to be a dominance in power relations, in which the highest power is held by BKKPN Kupang. However, BKKPN’s resources and mandate were limited. In principle, multiple services across landscapes should be levied by an equal scale of governing authorities, to maintain the resilience of ES.

Subsequently, the stakeholder valuation exercise identified the key ES, i.e., marine tourism, capture fisheries, supporting services (biodiversity benefit), and regulating services (coastal protection). Based on the stakeholders’ perception, ecosystem services in the highest criticality (importance—vulnerability) position is marine tourism, i.e., diving and glass-bottom cruise & snorkeling, and capture fisheries. The stakeholders hold a high priority (dependence—preference) for essential ecosystem services (regulating and supporting services) more than profitable services (marine tourism, capture fisheries). However, all things considered, the expectation for tourism revitalisation was still high. Overall, the stakeholder has a positive mindset towards promoting better ecological outcomes through good utilisation practices.

The spatial mapping of the coral reef ES further revealed spatial overlap between key ES delivery and utilisation areas, such as between marine tourism and capture fisheries and between capture fisheries and the biodiversity benefit/sanctuary zone. These overlaps can be perceived as pressuring interactions between each ES, as well as sources of trade-offs. Further translated into an ES interactional diagram, this study presumes that the combination of direct pressures (marine tourism and capture fisheries) and external pressure (land activities) act as negative drivers to the regulating services. As a result, the generated compounding pressure from the regulating and supporting services will affect the pressure source itself, forming a vicious cycle of value decline. Additionally, the economic valuation of each service, provided from previous studies, inferred that the services that receive compounding pressure are services with higher economic values than the activities that lead to direct or external pressure being applied to the ecosystem. It is inferred that exploitative activities should be leveraged to a scale, allowing for ecosystem recovery.

The evidence from this study provides communicable knowledge to the managing authorities and stakeholders to promote conjoined efforts towards better ecological outcomes. For further research avenues, this study could be accompanied by a thorough economic appraisal, to better identify the scope of intervention needed in each ES locus that can generate maximum benefit and minimize trade-offs.

## Figures and Tables

**Figure 1 ijerph-20-00089-f001:**
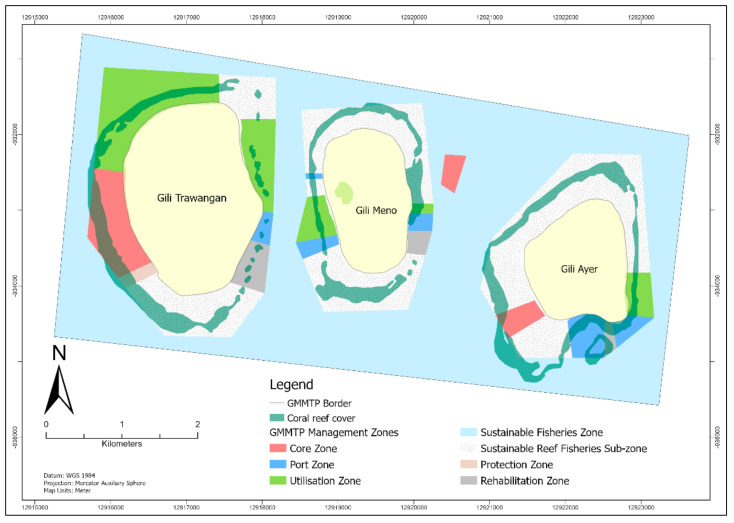
Map of Gili Matra Marine Tourism Park zoning system (Adapted and Modified from [11].)

**Figure 2 ijerph-20-00089-f002:**
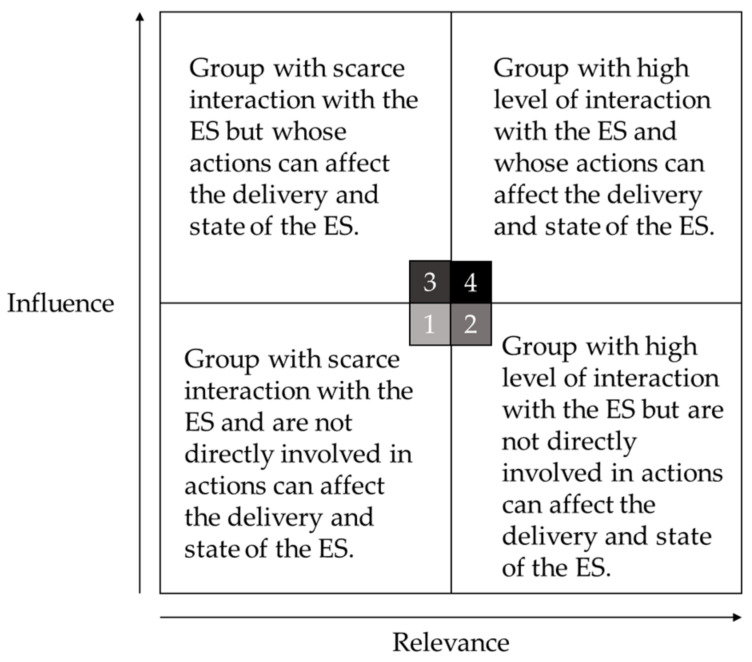
Different typologies of the stakeholder groups based on their relevance and influence over the ES (Adapted and modified from Figure 3 in [1]).

**Figure 3 ijerph-20-00089-f003:**
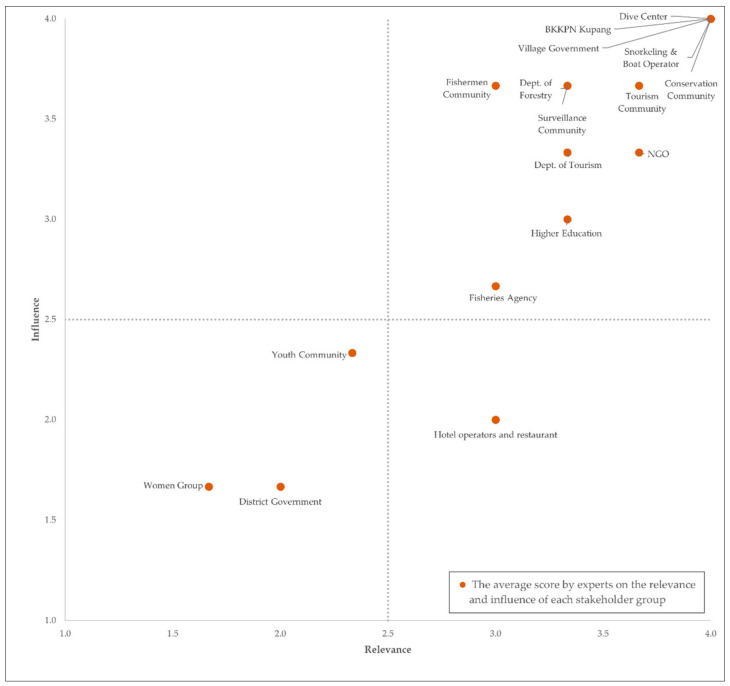
Assessment result of the local stakeholder groups of GMMTP. The “key stakeholders” are indicated by high influence (Y Axis) and high relevance (X Axis) values (highlighted in the upper-right quadrant).

**Figure 4 ijerph-20-00089-f004:**
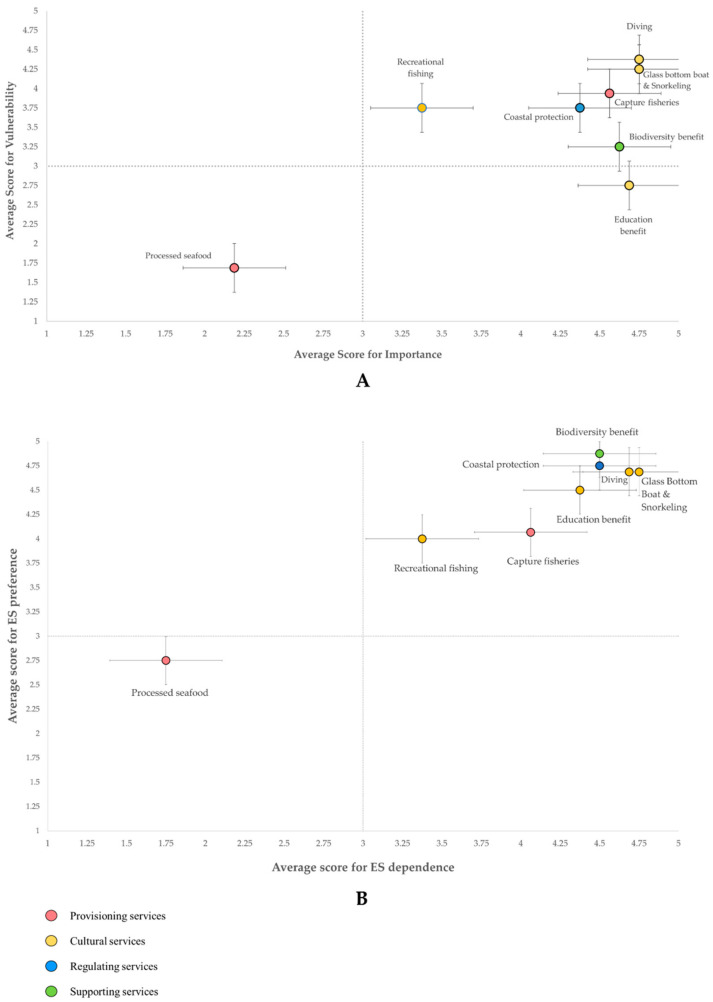
Four quadrant matrixes depicting stakeholder’s valuation for (**A**) Ecosystem Services criticality, and (**B**) Ecosystem Services priority; in the coral reef ecosystem of the GMMTP.

**Figure 5 ijerph-20-00089-f005:**
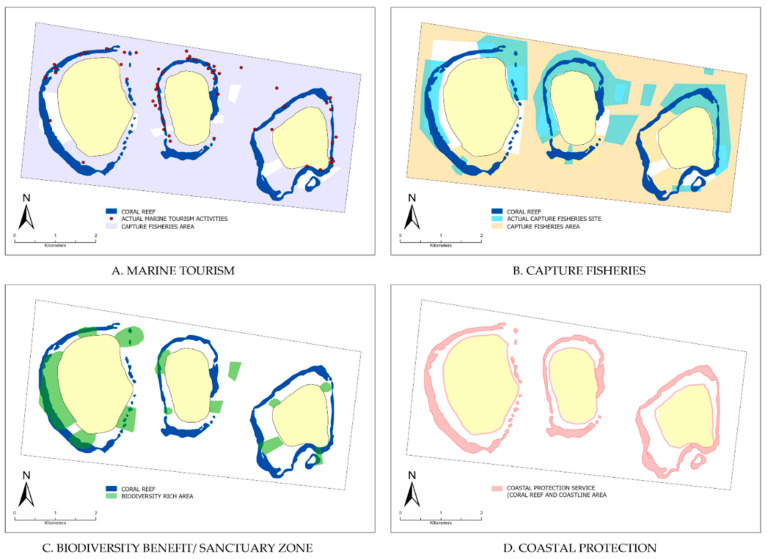
The location of key ecosystem services at the GMMTP coral reef: (**A**) Marine Tourism; (**B**) Capture Fisheries; (**C**) Biodiversity Benefit/Sanctuary Zone (Supporting Services), and; (**D**) Coastal Protection (Regulating Services).

**Figure 6 ijerph-20-00089-f006:**
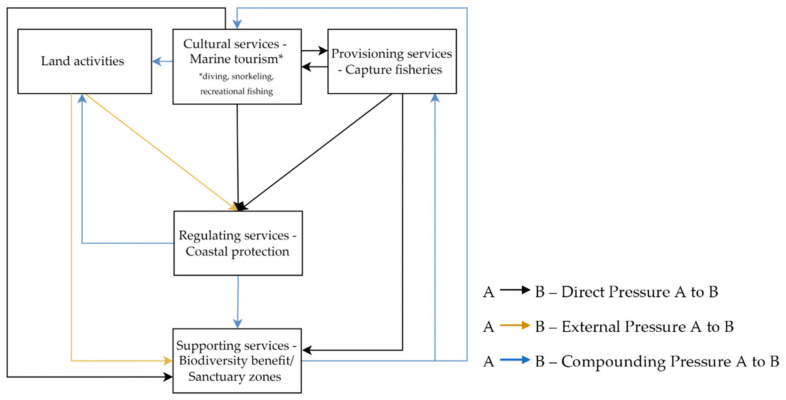
Conceptual diagram of the interaction of ecosystem services in the GMMTP coral reef ecosystem.

**Table 1 ijerph-20-00089-t001:** The Ecosystem Services and Stakeholders of the coral reef ecosystem within the Gili Matra Marine Tourism Park.

Categories	Service Type	Service Description	Stakeholders			Benefit Scale *
			Managing Agencies	Community	Private Business	
Provisioning Services	Fish and other species for food	Seafood (capture): Reef fishes Pelagic fishes Cephalopods	BKKPN Kupang, Fisheries Agency	Fishermen group, Conservation group, Surveillance group		Local
	Derivative Products	Processed Seafood Fish meatball Fish crisps Fish floss	BKKPN Kupang	Women Group		Local
Regulating Services	Coastal protection	Coastal stabilizer Erosion prevention Sediment accretion	BKKPN Kupang, Environmental Agency		Beach Tourism Operators (Hotel, restaurant, shop owners)	General
Supporting Services	Biodiversity benefit	Habitat and nursery for marine species Coral reef Reef fishes Sea turtles Sharks Giant clams	BKKPN Kupang, Environmental Agency	Surveillance group, Tourism group, Conservation group, NGO		General
Cultural Services	Recreation/Tourism	Recreational fishing	BKKPN Kupang, Tourism Agency	Fishermen group, Tourism group		Local
		Diving	BKKPN Kupang, Tourism Agency	Tourism group Conservation group	Dive operators	International
		Bottom Glass Boat & Snorkelling	BKKPN Kupang, Tourism Agency	Tourism group, Conservation group	Boat & Snorkelling operators	Local
	Educational benefit	Research & education benefit	BKKPN Kupang	Youth Group, Conservation group, NGO, Academia		Local

* existing & potential.

**Table 2 ijerph-20-00089-t002:** Role description of stakeholder groups for management of the coral reef ecosystem in the GMMTP [15].

Stakeholder	Role
BKKPN Kupang	Policy formulation, facilitation and monitoring of coastal and marine management
Dept. of Tourism	Policy formulation, facilitation and monitoring of tourism management
Dept. of Forestry	Policy formulation, facilitation and monitoring of environmental management
Dept. of Fisheries	Policy formulation, facilitation and monitoring of fisheries management
Village Government	Local administrator, facilitation for community to participate in environmental management
Dive Centre	Providing underwater tourism activity (scuba diving)
Snorkelling & Glass-bottom boat operator	Providing underwater tourism activity (snorkelling) and glass-bottom boat cruise
Tourism Community	Organising and increasing local community initiatives and participation on tourism activities
Conservation Community	Organising and increasing local community initiatives and participation on conservation efforts
Surveillance Community	Organising and increasing local community initiatives and participation on monitoring for zoning compliance
Fishermen Community	Organising and increasing the role of local fishermen on fisheries and conservation activities
NGO	Assist, mediate and advocate for good environmental management
Higher education/Academia	Conduct research for community services in ecosystem management

## Data Availability

Not applicable.

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
