# Peer review of "Assessment of Stakeholder’s Perceptions of the Value of Coral Reef Ecosystem Services: The Case of Gili Matra Marine Tourism Park"

_ijerph, 2022, doi:10.3390/ijerph20010089_

Round 1
Reviewer 1 Report
This manuscript describes a stakeholder perception study at Gili Matra Marine Tourism Park (GMMTP). The cover letter indicated the submission as a review article. However, this appears to be a research article https://www.mdpi.com/journal/ijerph/instructions)? Overall, the manuscript is well-written, although it is appearing more like a report document. Generally, throughout the manuscript, while relevant references were cited, they were almost always inserted at the end of the sentence, making it hard to figure out the relevance in relation to the research. More work needs to be done in this regard, e.g., “Following methodology applied in reference xxx… “ vs “In agreement with reference xxx, …” vs “Using data from reference xxx…”. Under Conclusions, a paragraph discussing the significance of the research findings on social well-being (linking ecosystem to public health) and to advancing knowledge in the field of the research, as well as, suggesting future research avenues will be helpful. Sharing a few other suggestions:
Ln 151-153: As it is, it is not clear where it is the authors that carried out this part of the research within a broader project. With reference 26 being in Indonesian, suggest providing a clearer explanation.
Ln 153-155: Suggest clarifying this process. How were the experts selected? How many and what were their expertise?
Figure 2: Is this a figure directly “extracted” from reference 28? Or was it adapted based on reference 28?
Ln172-174: Suggest differentiating “experts” and “stakeholders” so that readers can better follow the manuscript. This applies to the rest of the manuscript. It would be helpful if clearer explanation can be provided on the interview process. Were there two sets of interviews – carried under section 2.2.1 and also 2.2.2 (Ln191)? Suggest also indicating the availability of the supplemental material as an appendix.
Ln 181: From what I understood, the main stakeholders were characterised into these four categories, while the key ones under Category 4 will be the focal points? Suggest using a different word to differentiate core/main stakeholders vs key stakeholders identified in category 4. Suggest rewording this sentence to make this clear.
Ln310-314: Perhaps further elaborate on this “unexpected” finding under Discussions?
Ln338: Please provide a brief description of how were the 16 people selected and what were their background/relation/role to GMMTP?
Author Response
Thank you very much for your review. We are happy to take the journal's advice on whether the type of paper description for this manuscript should be changed in the journal's publication system. Please see the attachment for the detailed response on the review points.

Reviewer 2 Report
Overall, this paper demonstrates an interesting approach to identify key stakeholders and ecosystem services that is broadly transferable. My biggest concern is that I found the primary synthesis of the results (described in lines 224-229), including the maps and the conceptual diagram to not be “easily understandable” but instead confusing and inconsistent in the way terminology was used and the “management implementation outcomes” in places to be overly speculative or too generic. I think these issues can largely be resolved with some clarifications and being more careful and precise in the way terms are defined and described.
Line 60-62 – are the “studies” referred to the same as [14-17]? Because it is a separate paragraph, not clear what studies are being referred to here
Table 1 – it is not quite clear how “Benefit Scale” is determined; For example, Diving is the only one considered international? And I’m not sure what “General” means.
Line 153 – please briefly describe who these experts were (e.g., scientists, local residents, local businesses? GMMTP reef managers, etc), and roughly how many;
Line 173 – Was this on a scale of 1-5 or 1-4? And what generally were the questions, particularly of the line =2.5 dividing the quadrants? Presumably <3 not important, >3 important? I’m not seeing this question in the supplemental document (I see the ecosystem services likert questions, but not the stakeholder ones). I ask because if, for example, instead 1= something like “somewhat important” you could get a lot more variability in how the survey responders interpret that, and a cutoff of 2.5 becomes somewhat more arbitrary;
Line 191 – This was a likert scale again? You say this in results (line 345) , but some of that should be moved here to methods
Line 197 – How were the two combined? Scoring as “key” (upper right quadrant) in both matrices? Which would be the same I guess as scoring in the upper half for all 4 categories.
Line 225 – More details needed here on methods on
Line 261 – not sure what is meant by smaller scale here (i.e., spatial, temporal, intensity);
Line 273 – Do you mean these activities benefit from coastal protection, or do you mean that these activities can create impacts on coral reefs that impact coastal protection?
Line 274-276 – I’m a little confused by the wording here; First sentence says they were excluded from the ES identification and valuation; second sentence says the were included in the analysis; I think I’m not quite understanding the distinction; Or you mean they were “initially” excluded from the literature review, but based on discussions with experts you decided to add them in? But if you did add them back in, then why aren’t they in Table 1 as a “Tourism service”; Or you mean you excluded them as a “Tourism service” (because that would be an indirect benefit of coral reef services) but included them as stakeholders of “Coastal protection service” (because that would be a direct benefit)?
Section 3.1.1 – following on previous comment; I did have a little bit of trouble distinguishing what parts of this section were derived from literature review vs. expert discussion; Table 1 the combined final list based on both experts and literature? I think just a little bit of additional information on what ways the expert discussion modified what the literature review would be helpful
Line 308 – here you say key categories (plural), but methods defined category 4 as the key category (singular); And the stakeholders described (lines 309-310) are not Key or category 4, so confusing here; Maybe you mean “not found in the key category”?
Line 311 – “ranked low” - I think ‘lowly’ not used correctly here (adjective vs. adverb) according to google – I wasn’t sure myself!
Figure 3 – Was this on a 1-4 scale or 1-5 scale? If 1 to 4, I think just drop the “4.5” from the axes so its clear the upper right dot is the highest possible value; Might be interesting to show the expert error bars on these as well (if it doesn’t get too messy); These are averages (and among how many experts)? Add that to Legend (and methods).
Figure 4 – What are the error bars here? And why do they extend past an upper score of 5? Given a pretty small sample size, it might be interested to show the min/max scores for the error bars (if it doesn’t get too messy)
Line 393 – unclear what “it” refers to here; I think “coral reef” if this is a continuation of previous paragraph; otherwise if a new paragraph redefine “it”
Line 397 – Did you consider breaking down your list of ecosystem services into finer detail (e.g., turtle biodiversity, fish biodiversity, coral biodiversity); How the survey responders interpret the ES categories presumably would influence how they score their importance;
Line 399 – I don’t think this Appendix was included?
Line 434 – I think you need to be pretty precise about what exactly you are mapping here; You’re not really mapping the ES, you are mapping the areas where the ES are managed and/or interact with or be impacted by people;
Line 448 – is “traditional fishing” the same as “recreational fishing” [see Line 438]
Figure 5 –
i) Text here is small and blurry and very hard to read;
ii) Again you need to be precise about what you are mapping here as described in the legends and Figure legend. These really aren’t the “locations of key ecosystem services” – presumably the areas with diving and snorkeling also have good biodiversity? I think you are more specifically talking about the locations where these ES are managed or most likely to interact/be impacted by people;
iii) it is confusing that you use supporting & regulating for some, but the more detailed categories for others; suggest changing supporting to biodiversity and regulating to shoreline protection;
iv) Figure 5b- the light blue might should be labelled “traditional fishing areas” to better match the text (line 448)
v) 5c - This just appears to show the whole island is land? Or do you specifically mean the narrow perimeter around the whole island? Would it not be better to map the locations of key tourism areas (or are their literally hotels and restaurants over every square meter of land?) I’m honestly not sure why this is included, as it is not one of the “Key” ES identified in Figure 4, was not included in Table 1, is not an informative map, and the most important aspect of it (coastal protection) is otherwise mapped in Figure 5e
vi) 5d – suggest replacing “supporting services” with “dedicated sanctuary zones”, or similar
vii) 5e – again, I’m not sure the point of this one; It just seems to show the entire coastline;
Line 473-482 – Again I’m unclear on precisely what is being mapped here; The definition seems to change somewhat from map to map; Tourism and fishing seem to be most about where people obtain these services; Biodiversity seem to be where biodiversity is managed/protected; But land activities and coastline seem to be more about where land-based activities would negatively impact the reef (through erosion and coastal development) rather than where these benefits are received or managed? I would have thought for example, to see perhaps low elevation on land or zoning such as locations on land with high development where coastal protection would be most valued, or areas on land zoned for restaurants/hotels that would benefit from coastal protection; The degree to which these high tourism areas (for example) overlap with sanctuaries or fishing areas or snorkeling areas, etc. might be an interesting example of the ‘overlap’ mention in line 484; otherwise I’m struggling with how to interpret c and e
Line 478 – “the vital role of [land activities] in the provision of other coral reef ES”? I’m not sure what this means; I thought the primary benefit of reef ES to restaurants, hotels, etc. was through coastal protection (Line 273, Table 1)
Line 484 – “many overlapping spaces… that can be perceived as forces produced by mutually exclusive action” – I don’t understand what this means; Perhaps give an example from Fig. 5;
Line 486 – “an important challenge…” – would be helpful to give an example of what you mean here; presumably the conflict between fishing and tourism, and/or protected sanctuaries;
Line 506 - “entailed biodiversity loss” – awkwardly worded I think; coral reef cover and live coral cover wouldn’t entail biodiversity loss, would they?
Line 505 – This just seems like you are changing the interpretation of “regulating services” in Figure 6 to mean coral cover, live coral, coastal changes and development; It seems if you just said impacts to coastal protection in the figure, it would be more straightforward? Certainly there will be overlap between biodiversity, coastal protection, tourism, and fishing, so it seems somewhat awkward to try to parse them out in this way…
Line 510 – to ‘other biodiversity’ except that relevant to tourism and fishing? It seems unnecessary to me to try to parse out biodiversity in this way;
Figure 6 –
i) Again, this would be more straightforward if you specified Regulating services as Coastal protection and supporting services as Biodiversity;
ii) What happened to recreational fishing – is it considered under marine tourism?
iii) Why do land activities and marine tourism not affect supporting services?
iv) Why do land activities not affect marine tourism and fisheries?
v) I’m still struggling why land activities is singled out as both a beneficiary and pressure on regulating services, but tourism and fisheries (which also serve dual roles as stakeholders and pressures) are not; I wonder if it would be helpful to disentangle stakeholders of fisheries from the ecosystem service of fish, and stakeholders of tourism from tourism as an ecosystem service (e.g., aesthetic qualities, charismatic fauna, water clarity), in the same way land activities are defined as a stakeholder of coastal protection (in Table 1); See additional comment on Line 524;
vi) Does coastal protection only have indirect benefits to marine tourism and fisheries through changes in biodiversity? What about providing calm sheltered waters for boats or diving?
Line 519 – Why does tourism have a direct effect on regulating services (coral cover, live cover) but not biodiversity?
Line 524 – Then why isn’t there a black line from tourism to biodiversity? Should there also be a black arrow from marine tourism back onto itself (same for fisheries)? Or are you arguing that biodiversity supports tourism, tourism creates pressure on biodiversity? The specific way these boxes are being defined is unclear to me – tourism and fishing seem to have shifted to being pressures and/or beneficiaries of regulating and supporting services, rather than ecosystem services in and of themselves (as set up in Table 1);
Line 555 – Land activities were previously described as a tourism ecosystem service that indirectly benefits from reefs and in particular benefits from coastal protection (Table 1), but here instead they being identified as a pressure on coastal protection (but not other ecosystem services?); I just find this confusing and handled inconsistently;
Line 573 – “was … bidirectional” AND Line 589 “created compounding pressure” makes it sound like these are conclusions you are drawing, but it was your decision to diagram it this way (as a hypothesis of the underlying relationships), not as a conclusion from your analysis; I think saying “is” or “creates” or “was inferred to be” or “was diagrammed as” , “is assumed to” etc. would clarify this is something you are inferring not a conclusion you are drawing
Line 598 – “Rp10.821.883.500” the notation here and units seem different than how the other ecosystem services values were presented;
Line 635 – “hold a higher future expectation for essential services than profitable services, indicating a shfting focus to ecocentrism” - I’m not sure what is meant by this; “essential” vs. “profitable” have not previously been defined; biodiversity was listed as one of several essential (meaning “Key”) services on Line 289, but in Section 3.2 biodiversity was described as directly contributing to tourism and fisheries (which presumably are profitable); I also don’t understand from where you are deriving this temporal “shift” in focus or “future” expectation as the main thing mentioned in the results was wanting to revitalize the tourism economy;
Line 649 – “produced by mutually exclusive action”- not sure what is meant by this; if fishing areas and dive sites overlap – how is this interpreted as ‘mutually exclusive’? I also do not understand where or how coastal protection or land activities overlap with fishing, tourism, or biodiversity (based on Fig. 5 maps)
Line 649 –652 “posed as direct drivers to the regulating services” – you say this like it is a conclusion, but isn’t it something you are hypothesizing, or assuming, or inferring?
Line 650 – Doesn’t capture fisheries impact biodiversity too, not just regulating (Fig. 6)? And why wouldn’t marine tourism directly impact biodiversity?
Line 652 – “causes” seems to strong; you don’t have evidence of this, you are speculating or hypothesizing these relationships;
Line 653 – “simple calculations on the value of each service” – I don’t understand what this is referring to; The “value” estimates were Figure 4, right? Or do you mean the economic values described in Section 3.2?
Line 653 – “current trajectory” - I’m not sure what this is referring to either; You mean continuing to rely on fishing, if tourism doesn’t rebound; Or do you mean continued reliance on fishing and tourism is ultimately unsustainable because the reef will continue to degrade?
Line 654 – “loss in ecosystem essentials than yield from profitable services” – I’m not sure exactly what you mean here; Isn’t implementing management actions that protect and maintain biodiversity profitable (for tourism)?
Conclusions -Overall conclusions here seem a bit “doom and gloom” rather than using the maps and diagram to make positive management recommendations; most of the recommendations seem a bit generic (Line 615 – “manage ecosystem services”, Line 623 “include stakeholders in decision”) that don’t seem specifically derived from the analysis;
Author Response
Thank you very much for the detailed review. It has undoubtedly helped us to reflect on our research and seek improvement. Please see the attachment for the detailed response on the review points.

Round 2
Reviewer 2 Report
My prior review comments have been adequately addressed. I noticed a few minor things:
Line 327-328 This is an incomplete sentence, and I believe “hostels” should be “hotels”.
Line 367, Line 386, Line 398 – key stakeholder should be plural I think
Figure 5 – In the PDF, the image is hard to see overlaid on the old image, but it looks like the text is still hard to read and the letter key in the Figure legend may be wrong.
Author Response
Thank you for the reviews. We provide the additional revision as below:
- Do you mean lines 312- 314? The text revision is provided below:
“However, while conceptualising the ES interaction diagram, it was noted that land activities were closely associated with the coral reef’s function as a coastal protector; in specific, activities such as shoreline tourism developments where sand reclamation for beach leisure and hotel developments were flourishing [13,14].”
- Thank you, we have revised the nouns to “key stakeholders” in the lines below:
346: “Next, we conducted a stakeholder prioritisation exercise to find key stakeholders based on their relevancy…”
366: “Based on the prioritisation in Figure 3, key stakeholders include governmental agencies…”
377: “We conducted an ES appraisal with representatives of all 13 groups of identified key stakeholders (Table 2)…”
- We have revised the letter key in the figure legend/caption, and provided a clearer figure as attached in the table of responses.
